# Linking peak intensity of mechanically stimulated bioluminescence and cell surface area in dinoflagellates

Francis Letendre*, Abigail Blackburn and Michael Twardowski

## ABSTRACT

Mechanically stimulated bioluminescence (MSL) is present in most planktonic clades and marine ecosystems. The first flash kinetic parameters (FFKPs) and spectral properties are often species specific, making MSL a powerful tool for *in situ* ID and biodiversity assessments. The peak intensity (PI) of mechanically stimulated bioluminescence was measured for five species of dinoflagellates: *Alexandrium monilatum*, *Lingulodinium polyedra*, *Pyrocystis fusiformis*, *Pyrocystis noctiluca* and *Pyrodinium bahamense*. Peak intensity was assessed with respect to organism cell surface area and volume, building upon Seliger's rule, where previously a relationship was found linking cell surface area and total mechanically stimulated light (TMSL) (Buskey and Swift, 1990). These dinoflagellate species were chosen to cover a wide range of peak intensities ($10^8$ to $10^{10}$ photons/s) and surface area ($10^4$ to $10^6 \mu m^2$). Individual cells were isolated and individually photographed under a compound microscope, where cell size was measured. They were then dark-adapted and first flash emission from mechanical stimulation was measured with the Underwater Bioluminescence Assessment Tool (UBAT) from Seabird Scientific (www.seabird.com). Distributions of PI and surface area across all species were compared using non-parametric ANOVAs and a linear regression model, uncovering a strong positive correlation and strength of fit across all species between peak intensity and both cell surface area and volume. This study provides insight into understanding and potentially predicting the bioluminescence of organisms often responsible for significant primary and secondary productivity in marine waters with subsequent global impacts on fisheries and ecology. Bioluminescence measurements may also be a powerful tool for understanding plankton composition, ecology, and diversity.

KEY WORDS: Bioluminescence, Dinoflagellate, Surface area, UBAT

## INTRODUCTION

Bioluminescence is present in most marine systems and across all major marine phyla and can serve many functions, e.g. deterring predators, luring prey, finding mates in the water column, camouflage via counterillumination (Haddock et al., 2010). Recent reviews estimate bioluminescence evolved independently 0(100) times (Haddock et al., 2010; Lau and Oakley, 2021) and is most likely a post-Cambrian trait since interspecific uses of bioluminescence require visual predation. In the lab, marine bioluminescence can be triggered via light, electrical, chemical, and mechanical stimulation (Letendre et al., 2024a). However, apart from counterillumination or mate finding, naturally occurring bioluminescence in marine ecosystems is mainly stimulated mechanically.

In dinoflagellates, bioluminescence is emitted from scintillons, crystal-like organelles containing the luciferin substrate catalyzed by the luciferase enzyme (Fogel et al., 1972). Dinoflagellates produce their own type of luciferin (Haddock et al., 2010), which has been isolated in some krill species that feed on dinoflagellates. This suggests dinoflagellate predation may enable bioluminescent capabilities in other clades (Nakamura et al., 1989). Scintillons are most stable at a pH of 8.2, and bioluminescence output increases with lower pH, i.e. 6.5-7 (DeSa, 1964; DeSa and Hastings, 1968). Increasing the pH to 10 rapidly inactivates all bioluminescence capabilities of the organelles. The use and synthesis of luciferin follows a circadian rhythm in most dinoflagellate species, with emissions sometimes 60 times brighter during the night phase (Marcinko et al., 2013a,b; Sweeney, 1984).

Multiple parameters have been defined by the community to describe mechanically stimulated bioluminescence (MSL) and are grouped as first flash kinetics parameters (FFKPs). Mechanical stimulation is triggered when the membranes of cells or light organs are deformed by shear stress from turbulent kinetic energy induced by the feeding currents from an organism, although physical perturbations from swimming nekton, breaking surface waves, unstable internal waves, and turbulent interactions with benthic structures can also produce shears (Latz and Rohr, 1999, 2000; Latz et al., 2004). Once a shear threshold is reached, there is a latency of a few milliseconds – e.g. on average 17 ms for *Pyrocystis fusiformis* (Widder and Case, 1981a,b) and 15-22 ms for *Pyrodinium bahamense* (Latz et al., 2008) – before the emission starts, and rapidly reaches its peak intensity (Fig. 1A). This time from initial detection to peak intensity is usually described as the rise time (RT) (Fig. 1B). Then, instantaneous emissions decrease either exponentially or somewhat symmetrically back to zero. Peak intensities often increase with shear magnitude until an asymptote is reached, where higher shear does not induce higher flash intensities (Maldonado and Latz, 2007). Another metric used for quantifying MSL is total mechanically stimulable light (TMSL), which is described as the total amount of photons an organism can produce (photons/ind) (Letendre et al., 2024a). The individual is exposed to prolonged mechanical stimulation until exhaustion, i.e. bioluminescence is no longer possible due to luciferin/luciferase stock depletion.

Considering FFPKs are species dependent, potential applications for marine sciences are numerous. Previous efforts have used bioluminescence to monitor and detect early blooms of harmful algae and invasive species (Cusick and Widder, 2020), to estimate biomass and ecosystem productivity (Buskey, 1992; Lieberman et al., 1987; Neilson et al., 1995), to assess climate change induced

Harbor Branch Oceanographic Institute, Florida Atlantic University, Fort Pierce, Florida 32966, United States.

*Author for correspondence (fletendre@fau.edu)

F.L., 0000-0002-8687-1899

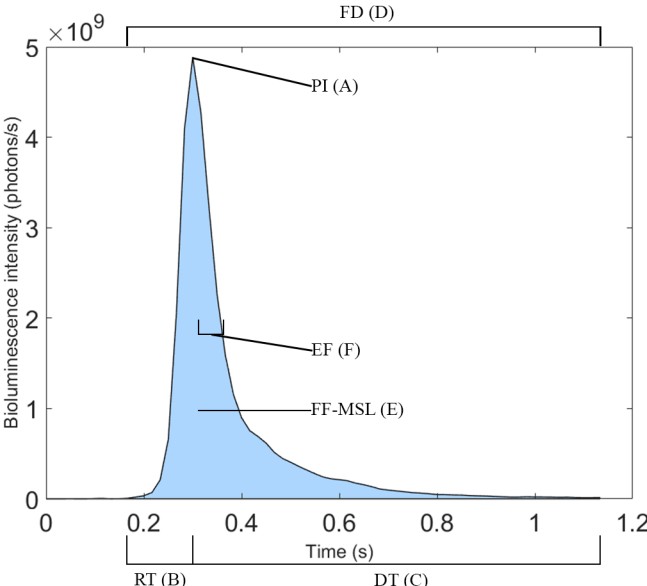

**Fig. 1. Typical first flash response of the dinoflagellate _P. noctiluca_ with annotated FFKPs.** This curve is averaged from the 53 first flash emissions used in Letendre et al. (2024b). Peak intensity (A), rise time (B), decay time (C), flash duration (D), first-flash mechanically stimulated light (E), E-folding time (F).

shifts in plankton communities (Piontkovski and Serikova, 2022), to study thin layer ecology (Widder et al., 1999), for flow visualization in laboratory (Foti et al., 2010; Rohr et al., 2002) and marine settings (Rohr et al., 1998; Stokes et al., 2004), and to identify fish school size and species composition (Altnağaç et al., 2010). Recently, first flash kinetics have been used to study diversity _in situ_, sometimes autonomously (Blackburn, 2023; Cronin et al., 2016; Johnsen et al., 2014; Letendre et al., 2024b; Messié et al., 2019). Some species, like _P. fusiformis_ (also studied here), can display two different flash forms upon mechanical stimulation; the first flash will display a typical exponential decay, whereas the second flash is more bell-shaped with a higher rise time (Widder and Case, 1981a,b).

Bioluminescence in dinoflagellates is primarily believed to prevent predation events by luring a second order predator, i.e. the burglar alarm hypothesis (Abrahams et al., 1993; Hanley and Widder, 2017; Porter and Porter, 1979). Another potential role is to act as a warning of the cell's toxicity to potential predators (aposematism) (Porter and Porter, 1979; Hanley and Widder, 2017). Indeed, in the presence of bioluminescent dinoflagellates, mortality rates of copepods are increased via predation from fish (Abrahams et al., 1993). Recent work has additionally found the physical disturbance from a predator fleeing a bioluminescent flash can also attract second order predators (Huang et al., 2024). Given that mean emitted wavelength maxima of dinoflagellate species usually vary between 470-480 nm (Letendre et al., 2024a), it is hypothesized that

blue-centered bioluminescence has evolved in dinoflagellates for high contrast with the organism's environment, and to closely match visual capabilities of potential predators (Oakley, 2025).

Typically, bioluminescence is measured _in situ_ using flow-through bathyphotometers (BPs) with a mixing chamber or a linear array of sensitive light detectors positioned after mechanical stimulation (Letendre et al., 2024a). Mixing chamber bathyphotometers used in the past include the Over-The-Side (OTiS) BP (Bivens et al., 2001), the NOSC BP (Latz and Rohr, 2013), BIOLITE (Latz and Rohr, 2013), the MBBP (Herren et al., 2005) and the Underwater Bioluminescence Assessment Tool (UBAT) currently sold commercially by Seabird Scientific. Depending on the science objectives, these types of BPs can be moored for autonomous monitoring, towed or profiled from a boat, and mounted on AUVs or gliders (Berge et al., 2012; Latz and Rohr, 2013; Moline et al., 2009; Shulman et al., 2020). For measurements of FFKPs under more controlled environmental conditions, the UBAT can also be used in lab settings given its compact size (Letendre et al., 2024b). Integrating spheres can additionally be used in the lab to measure TMSL or total chemically stimulated light (TCSL) (Buskey et al., 1992).

In the past, TMSL has been studied in relation to dinoflagellate cell surface area (Buskey et al., 1992). A consistent ratio, referred to as Seliger's rule, was found: $10^{11}$ photons/mm$^2$ (Buskey et al., 1992; Buskey, 1992). This study assesses an extension of Seliger's rule to cell surface area and the first flash peak intensity measured with the UBAT for five dinoflagellate species. If a consistent relationship is present in nature, then this could enable predictions of flash intensities using only the size distributions of bioluminescent plankton. This work continues exploration into assessing bioluminescence as a powerful tool for understanding plankton composition, ecology, and diversity.

## RESULTS

Mean equivalent spherical diameter (ESD), cell surface area (SA), cell volume (Vol) and peak intensities (PI) are compiled in Table 1. ESD ranged on average from 35.6 µm for _L. polyedra_ to 260 µm for _P. fusiformis_. It is worth noting that _P. fusiformis_ has an elongated pennate shape, whereas all other species here are nominally spherical (Fig. 2). SA ranged from $10^3$ to $10^3$µm$^2$. Vol followed the same trend across tested species, varying from $10^4$ to $10^4$µm$^3$. PI of the first flash varied from $10^8$ to $10^{10}$ photons/s. Distributions of PI all matched previously acquired data from (Letendre et al., 2024b) within the same order of magnitude. For the three variables, lowest PI, SA and Vol averages were attributed to _L. polyedra_, and highest to _P. fusiformis_ (Fig. 3A-C). First flash responses of the two _Pyrocystis_ species displayed sharp rise times and long exponential decays, while other tested species typically had a more symmetrical shape to their emissions. These bioluminescent first flash emission patterns were consistent with past studies (Latz et al., 2004; Letendre et al., 2024b). For all five species, PI had a much higher coefficient of variation (CV) when compared to their SA counterpart. Indeed, CV for PI varied from 40.7% for _L. polyedra_ to 76.1% for

**Table 1. Means and standard deviations of ESD, SA, Vol, and bioluminescent PI measured in the UBAT for the five tested species**

| Species (N) | ESD (µm) | SA (µm$^2$) | Vol (µm$^3$) | PI (photons/s) |
|---|---|---|---|---|
| _Pyrocystis fusiformis_ (33) | 260±17 | 2.65±0.308×10$^5$ | 1.29±0.224×10$^7$ | 1.59±1.01×10$^{10}$ |
| _Pyrocystis noctiluca_ (31) | 169±21 | 9.11±2.18×10$^4$ | 2.64±0.923×10$^6$ | 8.79±5.31×10$^9$ |
| _Alexandrium monilatum_ (44) | 51.7±6.2 | 8.53±2.00×10$^3$ | 7.56±2.64×10$^4$ | 3.98±3.00×10$^8$ |
| _Pyrodinium bahamense_ (30) | 40.2±3.2 | 5.11±0.799×10$^3$ | 3.47±0.791×10$^4$ | 2.93±1.58×10$^8$ |
| _Lingulodinium polyedra_ (30) | 35.6±4.1 | 4.03±0.890×10$^3$ | 2.45±0.779×10$^4$ | 1.11±0.451×10$^8$ |

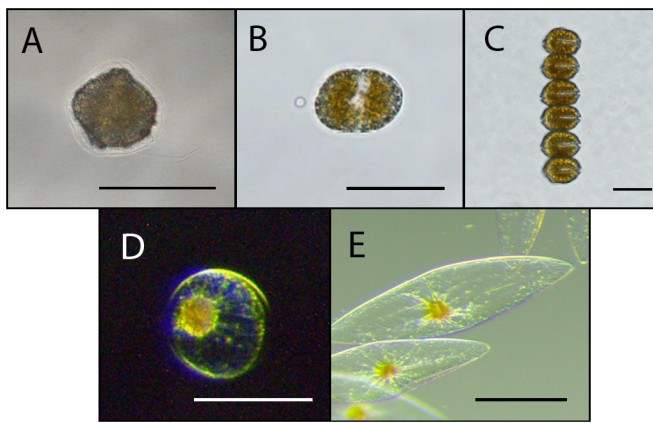

**Fig. 2. The five species of dinoflagellates tested in this study.** Sample size for each species is displayed in parentheses. (A) *Lingulodinium polyedra* (30). (B) *P. bahamense* (30). (C) *Alexandrium monilatum* (44). (D) *P. noctiluca* (31). (E) *P. fusiformis* (33). Scale bars are 50 um for A-C and 200 um for D,E.

*A. monilatum*. For SA, the lowest coefficient of variation was 11.7% for *P. fusiformis* and the highest 24.0% for *P. noctiluca*. Lower coefficients for SA could be due to all measured cells from a given species being sampled from the same generation. This was decided to reduce biases in bioluminescent emissions caused by intraspecific variation in organism development. Other causes of intraspecific variation are addressed in the discussion.

*Post hoc* ANOVA and the Kruskal–Wallis analyses found distributions of PI were statistically significant, apart from the two *Pyrocystis* species, possibly due to their taxonomic proximity. However, prior results from Letendre et al. (2024b) show PI as a distinguishable FFKP for these two species. Lower sample size and variables affecting FFKPs, e.g. cell stage, circadian rhythm, may contribute to differences in these results. Fig. 3A shows the respective distributions of the PI measured in the UBAT. All differences in distributions of SA and Vol were statistically significant ($P<0.05$), except for *P. bahamense* and *L. polyedra* (Fig. 3B,C). While these results establish differences in

bioluminescent emissions and cell size, another purpose here was to study this relation over a large range of cell sizes. This ensures relevance of the fitted curve and limits need for extrapolation (Fig. 4), since tested variables span three to four orders of magnitude, with *P. fusiformis* being one of the largest and brightest of all dinoflagellate species.

Fig. 4 displays first flash bioluminescence PI as a function of SA. Analysis of fit found the optimal regression model was a linear relation with no intercept ($R^2=0.55$). The scatter plot shows two distinct groupings, one consisting of the two *Pyrocystis* species and the other containing the three smaller species. A Pearson correlation test was made to assess the strength and significance of the relation ($\rho=0.75$, $P<0.05$). Significant correlation between SA and bioluminescence PI has implications regarding the placement of bioluminescent organelles within the cell, i.e. scintillons in dinoflagellates (see Discussion). It is important to note, however, that most of the *L. polyedra* cluster is below the linear fit, i.e. negative residuals. While the correlation is statistically significant, the surface area and PI relationship might prove less applicable for this particular clade. The linear slope of $6.16\times10^4$ photons $s^{-1}$ $\mu m^{-2}$, with 95% lower and upper confidence bounds of $5.51\times10^4$ and $6.82\times10^4$, respectively, allow to interpolate and estimate bioluminescence PI of dinoflagellate populations based on particle size distributions.

Fig. 5 shows the distribution of first flash PI ratioed to SA (photons $s^{-1}$ $\mu m^{-2}$) for each species. Following a Kruskal–Wallis test, the distributions of ratios were statistically indistinguishable for all species, with the exception of *L. polyedra* being distinct from both *Pyrocystis* species and *P. bahamense*. This distinction could be due to the narrower distribution of its ratios. Considering this caveat, the average ratio was $6.01\times10^4$ photons $s^{-1}$ $\mu m^{-2}$ for all species combined, closely matching the slope of the fitted curve in Fig. 4. Bootstrapping a two-tailed 95% confidence interval across the entire distribution for 1000 iterations showed an average ratio ranging from $5.18\times10^4$ to $6.87\times10^4$ photons $s^{-1}$ $\mu m^{-2}$. Converting the linear slope of Fig. 4 to $6.16\times10^{10}$ photons $s^{-1}$ $\mu m^{-2}$, it is possible to relate it to Seliger's rule for TMSL, i.e. $10^{11}$ photons $mm^{-2}$. Depending on an organism's bioluminescence capabilities – e.g. first-flash decay

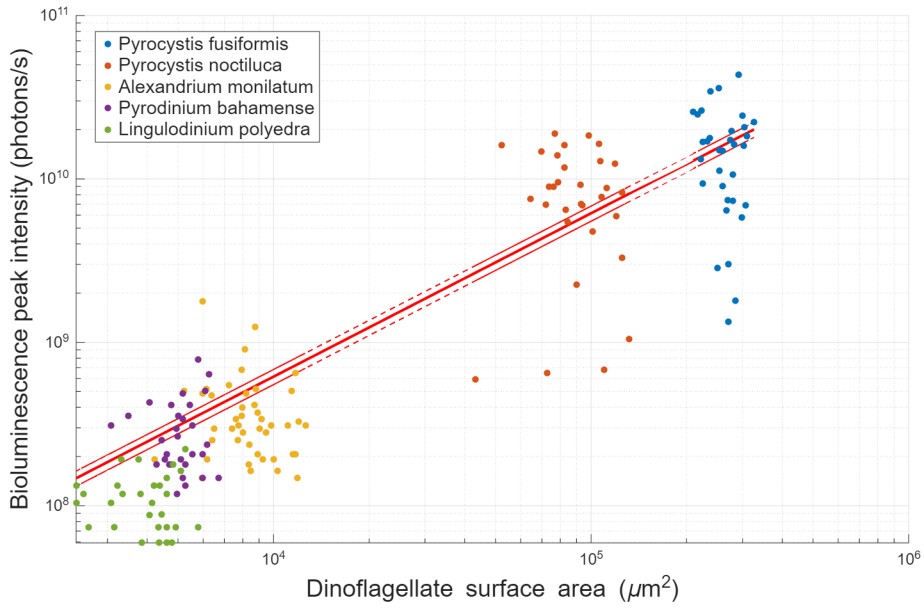

**Fig. 3. Scatter plot of bioluminescent PI as a function of SA.** Fitted curve is a linear regression without intercept, PI=$6.16\times10^4\times$SA. $R^2=0.55$. $N=168$. Red dashed lines represent upper and lower 95% confidence bounds of linear fit. The two distinct clusters correspond to large species (*Pyrocystis spp.*) and smaller species (*A. monilatum*, *L. polyedra* and *P. bahamense*). See Dataset 1 for the complete list of values.

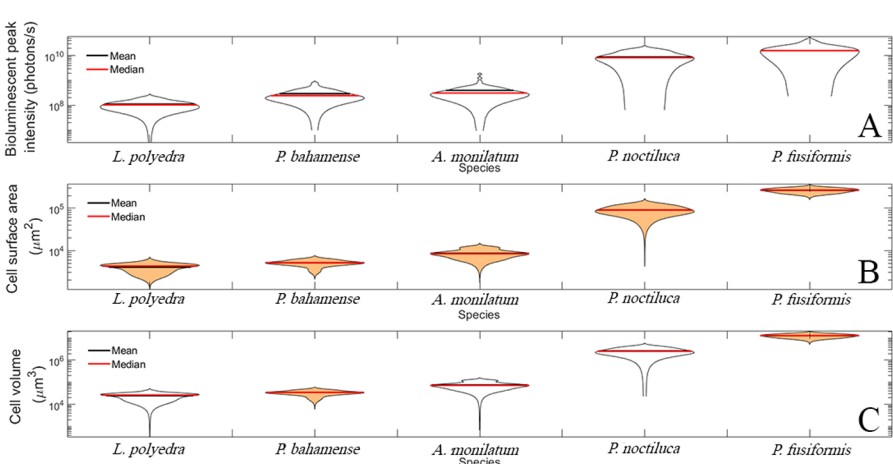

**Fig. 4. Violin plot of the bioluminescence PI (A), SA (B) and Vol (C) for all studied species.** Black and red lines indicate the distribution mean and median, respectively.

response, flash duration, number of flashes until luciferin stocks are depleted, refractory period – TMSL usually varies ± 1 order of magnitude relative to PI (Letendre et al., 2024a). For example, PI and TMSL of the dinoflagellate *N. scintillans* have been measured at $9.00 \times 10^{10}$ photons/s and $2.50 \times 10^{11}$ photons/ind, respectively (Buskey et al., 1992); for *Tripos fusus* (previously *Ceratium fusus*), PI and TMSL were measured at $3.41 \times 10^{9}$ photons/s and $5.30 \times 10^{8}$ photons/ind because this organism has a single, short flash (Buskey and Swift, 1990; Latz et al., 2004). Since spherical diameter, surface area and volume are perfectly correlated to one another, it is possible to derive the linear fit to these other variables. Thus, using ESD or cell Vol, PI can be estimated with $PI = 6.16 \times 10^{4} \pi \cdot ESD^{2}$ and $PI = 9.49 \times 10^{4} \pi \cdot Vol^{2/3}$, respectively.

## DISCUSSION
This present effort uncovered a strong and statistically significant positive correlation between dinoflagellate SA and mechanically stimulated bioluminescence PI ($\rho = 0.75$, $P < 0.05$). This linear fit allows for estimating bioluminescence output using ESD, surface

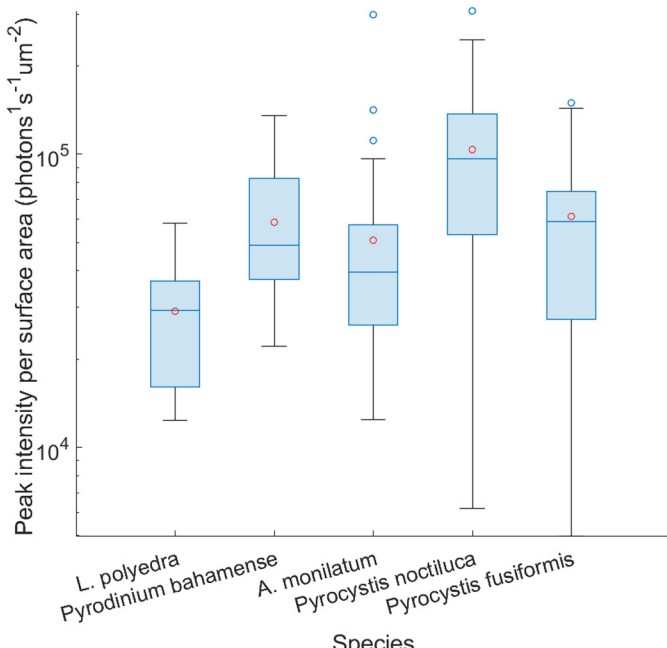

**Fig. 5. Boxplot of bioluminescent PI normalized to surface area.** Red circles indicate distribution means for each species.

area or Vol. For all five tested species, an average ratio of $5.18 \times 10^{4}$ to $6.87 \times 10^{4}$ photons $s^{-1}$ $\mu m^{-2}$ has been calculated (Fig. 5), while the fitted curve has a slope of $6.16 \times 10^{4}$ photons $s^{-1}$ $\mu m^{-2}$. This curve was fitted over a large range of dinoflagellate cell size, i.e. $10^{4}$ to $10^{6} \mu m^{2}$, for the model to be relevant across several dinoflagellate clades. These results provide insight into bioluminescence estimates in dinoflagellate-dominated systems, as well as harmful algal bloom monitoring.

### Broader context and future research directions
An important factor to take into account when comparing flash kinetics from different studies is the variation of shear stress among bathyphotometers. Indeed, based on instrument geometry and flow rate, each bathyphotometer has a probability function describing shear stress imposed on an organism as it passes through a grid, a constriction zone, or an impeller (Latz and Rohr, 2013). Since PI are directly correlated to shear stress, we *a priori* expect instrument-dependent responses (Maldonado and Latz, 2007). However, increasing shear stress past a species-specific threshold will result in emissions reaching an asymptotic region, where higher shear stress does not result in brighter flashes (Jing et al., 2011). Moreover, the asymptotic PI is dependent of the acceleration to which the organism is subject (Cussatlegras and Le Gal, 2007). Thus, unless a species-specific 'transfer functions' have been established, comparing the average ratio of $6.01 \times 10^{4}$ photons $s^{-1}$ $\mu m^{-2}$ to measurements with other bathyphotometers must be done with considerable care. For example, Latz and Rohr (2013) showed efficiency of stimulation (%TMSL) varied depending on species and instruments. Efficiency of stimulation is calculated as the ratio of an organism's integrated emission sampled in a bathyphotometer and its TMSL measured in an integrated sphere, giving the percentage of bioluminescence that can be stimulated from one pass in a given bathyphotometer. *P. fusiformis* efficiency was measured at 35 and 12% for the UBAT and the HIDEX-BP (a first flash, grid-stimulated bathyphotometer) (Latz and Rohr, 2013), respectively. However, the trend was inverted for *L. polyedra*, with 17 and 95% (Latz and Rohr, 2013). This could be due to the higher residence time of the UBAT, where cells could flash multiple times and increase their %TMSL, in comparison to the linear geometry of the HIDEX-BP. A multiple flash response of *P. fusiformis* has been well documented (Widder and Case, 1981a,b), whereas *L. polyedra* typically flashes once before a longer refractory period without flashing. Thus, the higher shear of HIDEX-BP may stimulate higher emission intensity in a single flash, resulting in higher %TMSL for species with long refractory periods. Comparing bioluminescence signals between bathyphotometers is a very complex process and

requires rigorous experiments to quantify shear stress ranges, critical flow rates, and controlling for multiple biotic and abiotic variables, e.g. light history, physiological state, temperature, salinity, organism size (Letendre et al., 2024a).

The geometry of the UBAT can also be responsible for intraspecific variation in measurements of bioluminescence PI. The UBAT is designed as a mixing flow-through bathyphotometer, where bioluminescence is sampled at 60 Hz within an integrating chamber (Orrico et al., 2009). By definition, an integrating sphere uniformly propagates radiant flux across a $4\pi$ surface. Therefore, flashing dinoflagellates are transformed into near perfect isotropic sources. However, prior calibrations and experiments with the UBAT showed up to 30% variation in intensity from a constant emitter when varying emitter location within the UBAT chamber, i.e. when the proximity of emitter to the pmt detector changed (Blackburn, 2023; Letendre et al., 2024b). This creates a bias in PI measurements since organism location greatly impacts the sampled signal. Additionally, recent flow simulations of the UBAT s-shaped light baffle showed rate of strain (linearly related to shear stress by the dynamic viscosity of the fluid) capable of prestimulating 90% of *P. fusiformis* and 40% of *L. polyedra* cells (Thombs et al., 2024). This further increases variation of emissions sampled in the mixing chamber, as species like *P. fusiformis* have two distinct flash responses depending on first or subsequent stimulation (Widder and Case, 1981a,b). Rate of strain inside the mixing chamber is also variable, i.e. most organisms will experience 50 s$^{-1}$, but up to 45% will experience rate of strain of 200 s$^{-1}$ (Thombs et al., 2024). Shear stress is the defining variable for triggering and scaling of mechanically stimulated bioluminescence (Cussatlegras and Le Gal, 2007). A bathyphotometer with well-defined excitation, e.g. HIDEX-BP, has a more constrained range or shear stresses at the excitation location, which is expected to result in less variance in first flash kinetic parameters (Latz and Rohr, 2013; Letendre et al., 2024a; Widder et al., 2003). However, every bathyphotometer has a critical flow rate (CFR), above which organism emissions stop increasing in relation to increased shear stress (Latz and Rohr, 2013). This is further complicated by species-specific shear stress thresholds. For example, Latz et al. (2004) measured a range of 2-8x difference in thresholds between *P. fusiformis* and *L. polyedra* in laminar pipe flow. The CFR for the UBAT prototype, i.e. the MBBP, was estimated at 0.5 l s$^{-1}$. However, where the current operating flow rate of UBAT falls with respect to the CFR for a specific organism is unclear (Herren et al., 2005).

Bioluminescence emissions of dinoflagellates are also subject to intraspecific variation (Marcinko et al., 2013a,b). Causes for such variation in dinoflagellates include physiological state, circadian rhythms, ontogeny, light history, nutrient levels, environmental parameters, and diet for heterotrophic or mixotrophic species (Letendre et al., 2024a). Indeed, at dusk, mechanically stimulated light increases until it reaches a maximum output during the night. This can introduce significant biases into collected data if species are not tested at the same time each day (see Materials and Methods). (Sullivan, 2000) showed *P. noctiluca* reaches maximum TMSL around 2 h post-sunset. Care should be taken to limit light exposure to cells before measurements, since intensities as low as 0.01 μmol photons m$^{-2}$ s$^{-1}$ can photoinhibit bioluminescence emissions of dinoflagellates (Hamman et al., 1981; Li et al., 1996; Sullivan and Swift, 1994). However, higher PAR during culture incubation results in higher bioluminescence capabilities in *Ceratium fusus* (Sullivan and Swift, 1995). It is paramount to incubate isolated cells in the dark before testing, as effects from photoinhibition can be measured for up to 45 min for *Tripos fusus* (Sullivan and Swift, 1994) and 1 h for *Pyrocystis acuta* (Hamman

et al., 1981) after light exposure. (Sweeney, 1982a,b) observed variation in *P. fusiformis* TMSL across five developmental stages and rhythmicity in day-night measurements, which is why individuals were sampled from a single generation in this present experiment. Temperature of the water column also affect the bioluminescence output. The PI of the invasive ctenophore *Mnemiopsis leidyi* decreased 20-fold when water temperatures were increased from the optimal 22°C to 30°C (Mashukova and Tokarev, 2012). Using the UBAT, (Chen et al., 2023) found temperature and salinity, over a range of 25°C and 5 ppt respectively, to be negatively correlated with PI of the red *Noctiluca scintillans*. Thus, samples need to be tested in water matching incubating conditions to avoid increased variation in FFKPs and biases when comparing different species. For heterotrophic dinoflagellates *Protoperidinum spp.*, lower food availability resulted in lower PI (Buskey et al., 1994; Jeong and Latz, 1994; Latz and Jeong, 1996). On the other hand, varying food sources does not seem to affect TMSL of *N. scintillans*, i.e. TMSL is independent of growth rate (Buskey, 1995). For *Protoperidinium huberi*, bioluminescence capabilities stopped after a 48 h of starvation (Buskey et al., 1994). Sweeney (1981) observed no change in bioluminescence output from *Lingulodinium polyedra* under prolonged nutrient limitation. In this sense, more work is needed to assess variations in emissions from cultured and natural populations of obligatory autotrophic dinoflagellates, i.e. nutrient rich media versus oligotrophic oceanic conditions. Measuring bioluminescence is a complex problem, requiring controls of biological, physical and instrumental variables.

While care should be taken while comparing bioluminescence measurements from different instruments, Buskey et al. (1992) measured the ratio of TMSL per surface area of *N. scintillans* at $7.6\times10^{10}$ photons mm$^{-2}$ in an integrating sphere. Ratios of up to $1.64\times10^{12}$ photons mm$^{-2}$ were measured for some species of heterotrophic *Protoperidinium spp*. These high ratios could potentially be explained by their high TMSL, obtained through multiple flashes, while being relatively small in size, e.g. less than 100 μm diameter. However, our proposed ratio of $6.01\times10^{10}$ photons s$^{-1}$ mm$^{-2}$ controls for differences in the number of possible flashes and reduces interspecific variation, since only the PI of the first flash was measured.

Past studies looking at scintillons placement within dinoflagellate cytoplasm using electron microscopy and histology have noted preferential placement of scintillons in the periphery of the cell (Heimann et al., 2000; Sweeney, 1980). This is even more apparent in some species of *Pyrocystis spp.*, as their large size allows observations of the cytoplasm under compound microscopy. This particular species will have cellular content, including scintillons, migrate to the periphery of the cytoplasm during the scotophase, i.e. dark phase, and retreat near the nucleus during the photophase (Seo and Fritz, 2000; Sweeney, 1982a,b). This ensures light produced in the scintillons will have the shortest pathlengths out of the cell and minimizes risks of light scattering within the cell.

Recent work uncovered *Pyrocystis noctiluca* can increase its Vol up to six-fold to regulate its buoyancy and migrate through the water column (Larson et al., 2024). A six-fold increase in volume is bound to greatly alter the ratio of bioluminescence PI to SA. This could contribute to *P. noctiluca* having the largest range in both SA and ratio of PI per surface area of the species studied (Figs 3B and 5).

## Applications

These results have applications in interpreting intraspecific and interspecific variability in bioluminescence emissions. When the bioluminescent dinoflagellate community from a particular system is

known, its size distribution may be used to estimate bioluminescence outputs upon mechanical stimulation. Inversely, the instantaneous light measured gives information on dinoflagellate densities in the water column in terms of total surface area and volume, in cases where a single dinoflagellate species is dominant. However, in order to identify the dinoflagellate community assemblage simply based on bioluminescence emissions, several more FFKPs, i.e. flash duration, rise time, e-folding time and decay time, need to be considered in the analysis (Letendre et al., 2024b). With those same species, Letendre et al. (2024b) achieved an identification accuracy of 73% using bioluminescence FFKPs. In this sense, measuring instantaneous bioluminescence *in situ* could be a tractable method for detecting and monitoring the development of HABs. For example, the Indian River Lagoon, Florida, is subject to cyclically annual blooms of toxic dinoflagellates *P. bahamense* and *Alexandrium monilatum*, both capable of bioluminescence (Phlips et al., 2011; Schaefer et al., 2019). A simple moored system measuring bioluminescence could help monitor fluctuations in cell densities, similar to *in situ* imaging (Barua et al., 2023; Cowen and Guigand, 2008; Erickson et al., 2012) and flow cytometry (Thyssen et al., 2015). In high diversity systems, where organism size distributions and densities span multiple orders of magnitude, smaller species, e.g. dinoflagellates, solitary radiolarians, may quickly have their bioluminescence estimates be shadowed by larger zooplankton at certain depths. In these cases, the relation proposed here would be most appropriate in upper layers of the water column, where dinoflagellate populations are concentrated.

The relation between bioluminescence PI and surface area should be investigated for larger zooplankton species, e.g. krill, copepods, and ctenophores. However, estimating surface area of these complex organisms could prove difficult. In addition, krill, copepods, and squids typically package luciferin and luciferase in specialized organs, i.e. photophores, which are usually precisely located on their ventral side (Young et al., 1980), and bioluminescence is mainly located along the meridional canals in ctenophores (Haddock and Case, 1999). In this sense, surface area may not be an appropriate variable for estimating bioluminescence in counterilluminaters (Johnsen et al., 2004).

The relation between dinoflagellate SA and PI can also be used in water column photon budget estimates. Indeed, instead of using an average emission for a single species, the relation can be used, coupled with measured particle distributions of the bioluminescent phytoplankton community, to account for intraspecific size variation. Photon budgets are complex analyses joining bioluminescence measurements at the individual scale and water column density counts (Batchelder and Swift, 1989; Batchelder et al., 1992). With dinoflagellates being responsible for about 30% of the bioluminescence in the first 40 m in Kongsfjord, Norway (*Protoperidinium spp.*) (Cronin et al., 2016), and the first 150 m of the Northern Sargasso Sea (*P. noctiluca*) (Batchelder et al., 1992), applying the proposed linear relation increase accuracy in surface layers and coastal estimates.

## MATERIALS AND METHODS
### Culture maintenance and species provenance
*A. monilatum* and *P. bahamense* were isolated from the Indian River Lagoon near Harbor Branch Oceanographic Institute (Florida, USA) using a 20 μm phytoplankton net (Fig. 2B,C). Cultures of *L. polyedra* (CCMP1738) and *P. noctiluca* (CCMP732) were acquired from The National Center for Marine Algae and Microbiota (NSCMA, Bigelow Laboratory, Maine, USA) (Fig. 2A,D). The starter culture of *P. fusiformis* was obtained from Pyrofarms (CA, USA) (Fig. 2E). Cultures were kept in a L1/2 - Si medium (Guillard and Morton, 2003) at 23°C and 38 ppt in a temperature controlled growth room and transferred every 2 weeks into fresh medium. These cultures have been kept in lab under these conditions for more than 2 years. To allow for

bioluminescent measurements during the day, cultures were illuminated with 80 μmol photons $m^{-2}$ $s^{-1}$ on a 12 h:12 h reverse day/night cycle.

### Bioluminescent measurements
Bioluminescence emissions were measured using the Underwater Bioluminescence Assessment Tool (UBAT, Seabird Scientific, www.seabird.com, Bellevue, WA, USA). The UBAT is a mixing chamber bathyphotometer, where organisms are pumped into a light baffle at 0.33 liters/s, before stimulation by the impeller upstream of the integration cavity (Orrico et al., 2009). Organisms are then mixed into a 440 ml integration cavity for up to 10 s (Orrico et al., 2009), where bioluminescence intensity (photons/s) is measured by a photomultiplier tube (PMT) at 60 Hz. Manufacturer calibration was independently verified with our integrating sphere assembly. The chamber is assumed to be fully turbulent (Thombs et al., 2024). For this particular experiment, the UBAT was placed in a covered black bin filled with temperature controlled artificial saltwater at 23°C and 38 ppt. A 20 μm Nitex mesh was placed at the output of the UBAT to prevent organisms from re-entering the UBAT and having secondary flashes sampled.

Individual cells from the five species were isolated, numbered and placed in 2.5 ml centrifuge tube caps using a Leica 205c stereomicroscope. Cells were picked from one generation to control for age as much as possible. Dinoflagellate cells were then photographed with a Nikon DS-Ri2 camera mounted on a Nikon Eclipse Ni-U. Tube caps containing single cells were then placed in a dark room for dark-adaptation and resting for 3 h. This resting period removed biases in light history and any prestimulation, ensuring the bioluminescent emission measured in the UBAT to be a first flash response. A black tarp covered the bin containing the UBAT for additional light-proofing. Cells were introduced by gently immersing the tube cap near the UBAT's flow intake. The subsequent cell sample was not introduced until a bioluminescent signal was detected by the PMT. This procedure allowed tracking of individual bioluminescent flashes with respect to cells of a known diameter. At least 30 bioluminescent signals for individual organisms per species were measured. A different species was measured each day at approximately the same time, to minimize effects of circadian rhythms on bioluminescent outputs.

### Data processing
Using the pictures taken under the Nikon compound microscope, the length and width of all cells were individually measured using the image processing software ImageJ. An ESD was calculated (Latz et al., 2004):

$$ESD = (l \times w^2)^{1/3}, \qquad (1)$$

where $l$ and $w$ are length and width, respectively. From the ESD, surface area $\left(4\pi\left(\frac{ESD}{2}\right)^2\right)$ and volume $\left(\frac{4}{3}\pi\left(\frac{ESD}{2}\right)^3\right)$ were calculated assuming sphericity.

Bioluminescent emissions acquired with the UBAT were processed and analyzed in MATLAB, including all statistical analyses. The 'findpeak' function was used to identify the highest intensity from each emission. All species datasets were inspected and individually processed with the 'rmoutliers' function, removing outliers and their corresponding variables that were three standard deviation above their respective means. PI three standard deviations lower were assumed to be unhealthy cells, and those higher were assumed to be most likely newly divided and/or aggregated cells during the dark adaptation process. Normality checks were carried out using the Lilliefors test. All distributions of SA and volumes agreed to a normal distribution within 95% confidence. For distributions of bioluminescent PI, normality was established with both species of *Pyrocystis* only. Following these checks, a one-way ANOVA test was performed on SA and Vol, and a Kruskal–Wallis was applied to PI data ($\alpha=0.05$). Since multiple comparisons of groups were done, the Bonferroni correction was applied to P-values to reduce Type 1 error probability (Dataset 2 contains the MatLab code for data processing).

### Acknowledgements
The authors sincerely thank Michael I. Latz and Malcolm Mcfarland for their initial guidance on culturing bioluminescent dinoflagellates and for providing starter cultures, Ed Malkiel for his help with calibration procedures, and Christopher Strait for lab coordination and help in data processing and visualization. We also thank the HBOI facilities for their support and assistance in this work.

## Competing interests
The authors declare no competing or financial interests.

## Author contributions
Conceptualization: F.L., A.B.; Formal analysis: F.L.; Funding acquisition: M.T.; Investigation: F.L.; Methodology: F.L., A.B.; Project administration: M.T.; Resources: M.T.; Supervision: M.T.; Validation: F.L., M.T.; Writing – original draft: F.L., A.B.; Writing – review & editing: F.L., A.B., M.T.

## Funding
Funding was provided by the Office of Naval Research (ONR) and the Postdoctoral Research Enhancement Program to Promote Excellence (PROPEL) of Florida Atlantic University. Open Access funding provided by Florida Atlantic University. Deposited in PMC for immediate release.

## Data and resource availability
Data is available upon demand to the corresponding author. All relevant data and details of resources can be found within the article and its supplementary information.

## Peer review history
The peer review history is available online at https://journals.biologists.com/bio/lookup/doi/10.1242/bio.062190.reviewer-comments.pdf

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
