## [Peer Review File · Biology Open]

Linking peak intensity of mechanically stimulated bioluminescence and cell surface area in dinoflagellates

Abigail Blackburn, Michael Twardowski and Francis Letendre
DOI: 10.1242/bio.062190

Editor: Lewis Halsey

Review timeline

Original submission:	28 July 2025
Editorial decision:	31 July 2025
Transfer to Biology Open:	31 July 2025
Editorial decision:	11 August 2025
First revision received:	18 August 2025
Accepted:	19 August 2025

Original submission

First decision letter

MS Title: Linking peak intensity of mechanically stimulated bioluminescence and cell surface area in dinoflagellates

Authors: Francis Letendre; Abigail Blackburn; Michael Twardowski
Article Type: Research Article

I have had the opportunity to read the above manuscript this evening. I found it to be a very interesting study, but unfortunately, the subject of your paper is not suitable for this journal, which is devoted to experimental comparative physiology, biomechanics and biochemistry of animals. On very rare occasions, we do publish articles on unicellular organisms and plants but there needs to be a strong link back to animals and the field of animal comparative physiology more broadly. I found this to be lacking in your article, substantiated from the paucity of comparative physiology journals that are cited in your manuscript. I am very sorry to give you this disappointing news and again apologise for not responding to your pre-submission enquiry in a timely manner.

I would like to stress that my decision not to send your paper out for full review is purely editorial and has nothing to do with the quality of your science.

Second decision letter

MS ID#: bio.062190

MS Title: Linking peak intensity of mechanically stimulated bioluminescence and cell surface area in dinoflagellates

Authors: Francis Letendre; Abigail Blackburn; Michael Twardowski

I have now reached a decision on the above manuscript.

The reviewer reports are shown at the bottom of this email or can be accessed, together with a copy of this decision letter, by going to:

As you will see, the reviewers raised a number of substantial criticisms that prevent me from accepting the paper at this stage.

They suggest, however, that a revised version might prove acceptable, if you can address their concerns. If you think that you can deal satisfactorily with the criticisms on revision, I would be pleased to see a revised manuscript. We would then return it to the reviewers.

At this stage, we also ask you to ensure your manuscript complies with our formatting guidelines. Provided you are able to fully address the referees' comments, we are positive about publication of your paper (we accept over 95% of revision submissions) and therefore hope you won't mind any extra work involved in reformatting your manuscript at this point.

Please ensure that you clearly highlight all changes made in the revised manuscript. Please avoid using 'Tracked changes' in Word files as these are lost in PDF conversion.

I should be grateful if you would also provide a point-by-point response detailing how you have dealt with the points raised by the reviewers in the 'Response to Reviewers' box. Please attend to all of the reviewers' comments. If you do not agree with any of their criticisms or suggestions please explain clearly why this is so.

Reviewer 1

This work focusses on the peak intensity of mechanically stimulated bioluminescence in five species of dinoflagellates that differ in surface area and volume. The goal is to provide further methodological evidence about using bioluminescence as a tool for understanding plankton composition, ecology, and diversity. Although this is a relevant topic, in its present form, it is not entirely clear what advancement is provided by the quantification of first flash emission using underwater bioluminescence assessment tool from Seabird Scientific.

1. From the results, this experimental setting appears to discriminate larger from smaller dinoflagellates, but it falls short when species of similar size are compared (see fig. 4). So how can we use this approach to study plankton composition?
2. The authors do not provide sensitivity analysis nor tests to quantify the error rate in post-hoc classification of single dinoflagellates. Given your models, which error do we have in classifying a species depending on its light emission?
3. It would be important to discuss the pros and cons of these methods when they are being applied to a water sample including a mix of different species in different proportions. What kind of information can we get? Might small be underestimated if the proportion of large species is near a certain value?

Reviewer 2

This manuscript presents an interesting study examining the relationship between peak intensity of mechanically stimulated bioluminescence and cell surface area in five dinoflagellate species. The work appears sound and the dataset spans a wide range of cell sizes and light intensities. The methods are clearly described, the figures are informative, and the results have potential applications in both fundamental plankton ecology and practical monitoring of harmful algal blooms.

However, I believe the manuscript could be strengthened through clearer framing, tighter linkage between results and discussion, and some streamlining of the statistical and interpretive approach.

Major points

1. The introduction is generally good but a few sentences are overly dense or ambiguous. For example, the reference to "currents produced by the luminescent organism" is unclear. Does this mean self-generated feeding currents or currents from another organism? These should be clarified.

2. Both one-way ANOVA and a non-parametric Kruskal-Wallis test are applied to the same datasets. Unless there is a specific reason to present both, I suggest selecting the most appropriate test for each dataset and explaining why it was chosen. Presenting both without justification seems redundant.

3. The regression of peak intensity against surface area shows that all *L. polyedra* points have negative residuals. This suggests that the relationship observed for other species may not apply equally to this taxon. This deviation should be explicitly addressed in the discussion, as it could have implications for generalizing the relationship across taxa.

4. I found the discussion to be the weakest part of the manuscript, mainly because it was extremely dense and difficult to identify the key take home messages. I also felt very disconnected from the results. The discussion currently begins with a dense technical section on shear stress and bathyphotometer geometry, without first summarizing the main findings. I recommend starting with a short, clear paragraph that states the key results (slope, correlation strength, interspecific differences) and their importance in relation to the goals of the study and more broadly.

5. Much of the discussion reads like a broad review of all possible sources of variation in dinoflagellate bioluminescence (e.g., temperature, salinity, food availability, nutrient status, photoinhibition, life stage). While these factors are relevant to the field, many were not measured or manipulated in the present study. Long digressions on these topics cause the discussion to feel disconnected from the actual results. This content could be condensed and reframed as "broader context and future research directions."

6. The strongest parts of the discussion are where the authors tie their findings back to the main goals of the study, consider species-specific deviations, and explore applications to monitoring and photon budgets. These should be brought forward and mapped more tightly onto the flow presented in the results section.

7. The applied implications (e.g., predicting bioluminescence from size distributions, HAB monitoring) are compelling but currently scattered throughout the discussion. Consolidating them into a single, well-structured section would make the potential utility of the findings more apparent.

Minor points

- Figures are generally clear, but it might help to briefly note in the caption for Fig. 4 that the two distinct clusters correspond to large (*Pyrocystis*) vs. small species.
- Ensure all abbreviations are defined at first use.
- There are a few typographical and minor grammatical issues throughout the manuscript that could be addressed during revision.

Reviewer's Responses to Questions

Experimental quality

Does each figure have the proper controls?

If 'No', please indicate reasons in Comments for Author box below.

Reviewer #1:

- Yes

Reviewer #2:

- Yes

Were the data analyzed using appropriate statistical tests?

If 'No', please indicate reasons in Comments for Author box below.

Reviewer #1:

- Yes

Reviewer #2:

- Yes

Reproducibility

Were experiments performed using adequate number of biological replicates?

If 'No', please indicate reasons in Comments for Author box below.

Reviewer #1:

- Yes

Reviewer #2:

- Yes

Does the methods section provide sufficient detail to permit reproducibility?

If 'No', please indicate reasons in Comments for Author box below.

Reviewer #1:

- Yes

Reviewer #2:

- Yes

Completeness

Are the manuscript's conclusions supported by the data?

If 'No', please indicate reasons in Comments for Author box below.

Reviewer #1:

- Yes

Reviewer #2:

- Yes

Scholarship

Do the authors cite and discuss the merits of data that would argue for and against their conclusion?

If 'No', please indicate reasons in Comments for Author box below.

Reviewer #1:

- Yes

Reviewer #2:

- Yes

Does the manuscript title & abstract accurately reflect the contents of the manuscript, without hyperbole?

If 'No', please indicate reasons in Comments for Author box below.

Reviewer #1:

- Yes

Reviewer #2:

- Yes

First revision

Author response to reviewers' comments

Reviewer 1: This work focusses on the peak intensity of mechanically stimulated bioluminescence in five species of dinoflagellates that differ in surface area and volume. The goal is to provide further methodological evidence about using bioluminescence as a tool for understanding plankton composition, ecology, and diversity. Although this is a relevant topic, in its present form, it is not entirely clear what advancement is provided by the quantification of first flash emission using underwater bioluminescence assessment tool from Seabird Scientific.

1. From the results, this experimental setting appears to discriminate larger from smaller dinoflagellates, but it falls short when species of similar size are compared (see fig. 4). So how can we use this approach to study plankton composition?

This comment has been addressed. Clarifications on the strength of fit on the *L. polyedra* cluster were added. Regarding plankton composition, the authors make the claim that when the dinoflagellate species assemblage is known, bioluminescence output can be estimated. To do this in reverse (ID species based on bioluminescence output), much more details are needed. The paper Letendre et al., 2024b explores this very question. Further clarifications were added in the discussion.

2. The authors do not provide sensitivity analysis nor tests to quantify the error rate in post-hoc classification of single dinoflagellates. Given your models, which error do we have in classifying a species depending on its light emission?

This comment has been addressed. In this study, we do not aim to identify species solely by their bioluminescence emission. Since the statistical analyses involve multiple group comparisons, the Bonferroni correction was newly applied to the dataset and p-values in the effort to compensate for type 1 error accumulation. This was added to the manuscript and clarified. To correctly identify species solely on bioluminescence emissions, other flash kinetics such as flash duration, rise time and decay time are needed, otherwise the error is too large. Letendre et al., 2024b described precisely this goal, as we aimed to identify these same 5 species using their flash kinetics. Using FFKPs, the model was 73% accurate in correctly identifying the dinoflagellate species. However, for this present study, the goal was not to identify species based on peak intensities, but to estimate bioluminescence output when the community is known.

3. It would be important to discuss the pros and cons of these methods when they are being applied to a water sample including a mix of different species in different proportions. What kind of information can we get? Might small be underestimated if the proportion of large species is near a certain value?

This comment has been addressed. Nuances have been added throughout the discussion. Problems associated with a diverse water sample have been discussed and the potential for identification of species based on bioluminescence output has been tied to a previous publication.

Reviewer 2: This manuscript presents an interesting study examining the relationship between peak intensity of mechanically stimulated bioluminescence and cell surface area in five dinoflagellate species. The work appears sound and the dataset spans a wide range of cell sizes and light intensities. The methods are clearly described, the figures are informative, and the results have potential applications in both fundamental plankton ecology and practical monitoring of harmful algal blooms.

However, I believe the manuscript could be strengthened through clearer framing, tighter linkage between results and discussion, and some streamlining of the statistical and interpretive approach.

Major points

1. The introduction is generally good but a few sentences are overly dense or ambiguous. For example, the reference to "currents produced by the luminescent organism" is unclear. Does this mean self-generated feeding currents or currents from another organism? These should be clarified.

This comment has been addressed. Overly long sentences have been modified for better flow and reading.

2. Both one-way ANOVA and a non-parametric Kruskal-Wallis test are applied to the same datasets. Unless there is a specific reason to present both, I suggest selecting the most appropriate test for each dataset and explaining why it was chosen. Presenting both without justification seems redundant.

This comment has been addressed. The ANOVA test was applied to cell surface area and cell volume, whereas the Kruskal-Wallis was applied to the peak intensity. This was decided since the first two variables were normally distributed, contrary to the peak intensity variable. The text describing the analyses has been re-worded.

3. The regression of peak intensity against surface area shows that all *L. polyedra* points have negative residuals. This suggests that the relationship observed for other species may not apply equally to this taxon. This deviation should be explicitly addressed in the discussion, as it could have implications for generalizing the relationship across taxa.

This comment has been addressed. Details have been added regarding the negative residuals of this particular subdataset in the result section.

4. I found the discussion to be the weakest part of the manuscript, mainly because it was extremely dense and difficult to identify the key take home messages. I also felt very disconnected from the results. The discussion currently begins with a dense technical section on shear stress and bathyphotometer geometry, without first summarizing the main findings. I recommend starting with a short, clear paragraph that states the key results (slope, correlation strength, interspecific differences) and their importance in relation to the goals of the study and more broadly.

This comment has been addressed. A paragraph summarizing key results and stating the main applications has been added at the start of the discussion.

5. Much of the discussion reads like a broad review of all possible sources of variation in dinoflagellate bioluminescence (e.g., temperature, salinity, food availability, nutrient status, photoinhibition, life stage). While these factors are relevant to the field, many were not measured or manipulated in the present study. Long digressions on these topics cause the discussion to feel disconnected from the actual results. This content could be condensed and reframed as "broader context and future research directions."

This comment has been addressed. The discussion has been divided into two subsections: Broader context and future research directions and Applications. Content in the discussion has been moved and re-worded for better flow and clarity.

6. The strongest parts of the discussion are where the authors tie their findings back to the main goals of the study, consider species-specific deviations, and explore applications to monitoring and photon budgets. These should be brought forward and mapped more tightly onto the flow presented in the results section.

This comment has been addressed. Parts of the discussion have been streamlined and key points have been reinforced.

7. The applied implications (e.g., predicting bioluminescence from size distributions, HAB monitoring) are compelling but currently scattered throughout the discussion. Consolidating them into a single, well-structured section would make the potential utility of the findings more apparent.

This comment has been addressed. The applications of our current effort have been consolidated into the last two paragraphs of the discussion.

Minor points

- Figures are generally clear, but it might help to briefly note in the caption for Fig. 4 that the two distinct clusters correspond to large (Pyrocystis) vs. small species.

This comment has been addressed. The Figure 4 caption has been revised.

- Ensure all abbreviations are defined at first use.

This comment has been addressed. All abbreviations are defined at first use.

- There are a few typographical and minor grammatical issues throughout the manuscript that could be addressed during revision.

This comment has been addressed. The manuscript has been proofed for clarity and typographical errors.

Third decision letter

MS ID#: bio.062190R1

MS Title: Linking peak intensity of mechanically stimulated bioluminescence and cell surface area in dinoflagellates

Authors: Francis Letendre; Abigail Blackburn; Michael Twardowski

I've had the chance today to take a good look at your revised manuscript and responses to the reviewers. While they had a good number of comments, I can see that you and your colleagues have responded to them thoroughly. Therefore, I am happy to tell you that your manuscript has been accepted for publication in Biology Open, pending our standard publication integrity checks. It was accepted on 19th August 2025.